# The Paleochrono-1.1 probabilistic model to derive a common age model for several paleoclimatic sites using absolute and relative dating constraints

Frédéric Parrenin[1], Marie Bouchet[2], Christo Buizert[3], Emilie Capron[1], Ellen Corrick[4], Russell Drysdale[5], Kenji Kawamura[6], Amaëlle Landais[2], Robert Mulvaney[7], Ikumi Oyabu[6], Sune Olander Rasmussen[8]

[1]Univ. Grenoble Alpes, CNRS, INRAE, IRD, Grenoble INP, IGE, 38000 Grenoble, France
[2]UMR8212, CEA–CNRS–UVSQ–UPS, Laboratoire des Sciences du Climat et de l'Environnement (IPSL), Gif-sur-Yvette, France
[3]College of Earth, Ocean and Atmospheric Sciences, Oregon State University (OSU), Corvallis, OR, USA
[4]School of Earth, Atmosphere and Environment, Monash University, Clayton, Victoria, Australia
[5]School of Geography, The University of Melbourne, Melbourne,Victoria, Australia.
[6]National Institute of Polar Research, Research Organizations of Information and Systems, 10-3 Midori-cho, Tachikawa, Tokyo 190-8518, Japan
[7]British Antarctic Survey, Madingley Road, High Cross, Cambridge, CB3 0ET, UK
[8]Physics of Ice, Climate and Earth, Niels Bohr Institute, University of Copenhagen, Copenhagen, Denmark.

*Correspondence to*: Frédéric Parrenin (frederic.parrenin@univ-grenoble-alpes.fr)

**Abstract** Past climate and environmental changes can be reconstructed using paleoclimate archives such as ice cores, lake and marine sediment cores, speleothems, tree rings and corals. The dating of these natural archives is crucial for deciphering the temporal sequence of events and rates of change during past climate changes. It is also essential to provide quantified estimates of the absolute and relative errors associated with the inferred chronologies. However, this task is complex since it involves combining different dating approaches at different paleoclimatic sites and often on different types of archives. Here we present *Paleochrono-1.1*, a new probabilistic model to derive a common and optimised chronology for several paleoclimatic sites with potentially different types of archives. Paleochrono-1.1 is based on the inversion of an archiving model: a varying deposition rate (also named growth rate, sedimentation rate or accumulation rate) and also, for ice cores, a lock-in-depth of air (since, in the absence of significant surface melt, the air is trapped in the ice at about 50-120 m below the surface) and a thinning function (since glacier ice undergoes flow). Paleochrono-1.1 integrates several types of chronological information:

prior knowledge of the archiving process, independently dated horizons, depth intervals of known duration, undated stratigraphic links between records, and, for ice cores, Δdepth observations (depth differences between events recorded synchronously in the gas and solid phases of a certain core). The optimization is formulated as a least-squares problem, assuming that all probability densities are near-Gaussian and that the model is nearly linear in the vicinity of the best solution. Paleochrono-1.1 is the successor of IceChrono, which produces common and optimized chronologies for ice-cores. Paleochrono-1.1 outperforms IceChrono in terms of computational efficiency, ease of use, and accuracy. We demonstrate the ability of Paleochrono-1.1 in an experiment involving only the MSL speleothem in Hulu Cave (China) and compare the resulting age model with the SISALv2 age models. We then demonstrate the multi-archive capabilities of Paleochrono in a new ice-core–speleothem dating experiment, which combines the Antarctic Ice Core Chronology 2023 dating experiment, based on records from five polar ice cores, with data from two speleothems from Hulu Cave dated using uranium/thorium radiometric techniques. We analyse the performance of Paleochrono-1.1 in terms of computing time and memory usage in various dating experiments. Paleochrono-1.1 is freely available under the MIT open-source license.

# 1 Introduction

Numerous natural archives provide records of the past evolution of climate and environment. Continuous paleoclimate archives require a continuous deposition process. Examples are: ice cores (Dome Fuji Ice Core Project Members et al., 2017; EPICA community members et al., 2004; EPICA Community Members et al., 2006; NEEM community Members, 2013; NorthGRIP project members, 2004; WAIS Divide Project Members, 2013), marine sediment cores (Elderfield et al., 2012; Lisiecki and Raymo, 2005; Shackleton, 2000; Shackleton et al., 2000), lake sediment cores (Hodell et al., 1999; Williams et al., 1997) and speleothems (Cheng et al., 2018; Corrick et al., 2020; Wang et al., 2001). In the case of speleothems, however, we should note that the deposition (i.e., speleothem growth) sometimes is only episodic, that is, continuous only during some time intervals. Such archives can directly record or more indirectly reflect various climatic parameters, such as local atmospheric or oceanic temperature,

precipitation amount and seasonality, sedimentation or precipitation rate, atmospheric composition, sea level, ocean circulation intensity, insolation or biosphere activity.

For paleoclimate archives to provide precise insight into past climate change, it is a prerequisite to be able to derive the age of the archive at each depth level, i.e. to derive an associated chronology. This is, however, a complex task which involves different dating methods (Brauer et al., 2014), depending on the archive type and the temporal coverage. We will not go into the detail of each method for every archive, but we list here the different types of chronological information:

1) **Modelling of the archiving process:** it is often possible to model deposition through time. For an ice core, this is complicated by the fact that air bubbles are trapped at depth and not at the surface, hence the air is systematically younger than the surrounding ice by a non-constant amount. Moreover, ice layers thin with time due to ice flow. For some archives (e.g. speleothems), the sedimentation or growth rate may vary abruptly due to climatic or local environmental factors, making it a challenge to accurately describe the deposition/growth process.

2) **Dated horizons:** some horizons in the archive can be dated using an independent method. Examples include a volcanic ash layer that is dated either from historical records or by radiometric dating of the ash, a paleomagnetic or solar event identified by variations in the $^{10}$Be concentration or remanent magnetization, or a layer dated by radiometric analysis such as U/Th, $^{14}$C or $^{81}$Kr.

3) **Intervals of known duration:** sometimes, a section of an archive is of known duration (typically, a section from an ice core or tree where annual layers can be counted), although the absolute age of the section may not be known accurately.

4) **Stratigraphic links:** links can be derived between the depths of several paleoclimate sites that are known (or assumed) to have the same age. Typically, this can be the same volcanic event recognized at different sites, or an abrupt Dansgaard-Oeschger event seen in multiple proxy records which is assumed to be synchronously recorded within a given margin.

5) $\Delta_{depth}$ **observations:** can be seen as stratigraphic links between ice and air phases of a single ice-core site (Parrenin et al., 2012).

Each dating method and paleoclimate archive has strengths and weaknesses in terms of chronological construction. For example, ice cores have robust archiving models, which accurately describe the

85 processes of snow deposition, densification and ice flow (e.g., Parrenin et al., 2004, 2007) and at some site, the annual layers can be counted over long sections (e.g., Sigl et al., 2016; Svensson et al., 2008). However, ice cores generally do not have strong constraints on absolute ages beyond the last millenia. Speleothems can be dated very accurately and precisely using uranium-thorium (U-Th) dating methods (e.g., Cheng et al., 2018). However, speleothem records are often less continuous than ice or marine

sediment records due to hiatuses in calcite precipitation, which can be related to drier or colder conditions above the cave but may also be unrelated to climate (e.g. due to changes in the percolation path of the source waters). Marine sediment cores have the longest temporal extent (e.g., Lisiecki and Raymo, 2005). However, prior to the radiocarbon dating limit (~50 kyr BP, thousands of years before 1950 A.D.), their chronologies rely on tuning to different reference records such as those based on the calculation of Earth's

astronomical parameters (Laskar et al., 2004). In paleoclimatology, it is typical to treat individual records independently, making comparisons between chronologies difficult. Combining chronological data from multiple archives, across various archive types, has the potential to improve our chronological framework. An appropriate strategy is to combine the results from the different dating methods and paleoclimatic sites using a probabilistic framework in order to make the best out of the different archives and records.

A probabilistic framework was originally developed for polar ice cores using the Bayesian model *Datice* (Lemieux-Dudon et al., 2010b, 2015). Datice produces an optimal chronology using different types of chronological information from multiple ice cores. Using this method, the ice-core chronology is derived from an archiving model that relies on the three canonical glaciological quantities of the ice-core dating problem: the deposition rate also called accumulation rate, the lock-In depth (LID) of air and the thinning

function (i.e. ratio of the present-day thickness of an annual layer to its initial thickness when it was deposited at the surface). In Bayesian terminology, the initial estimate of this archive model is called the prior scenario.

Datice then seeks an optimal scenario, which is the best compromise between the prior scenario of the archiving process and the various chronological data. Datice was used in the construction of the reference

chronology Antarctic Ice Core Chronology 2012 (hereafter AICC2012), combining archiving models and chronological information from four Antarctic and one Greenland ice cores (Bazin et al., 2013; Veres

et al., 2013). Datice was entirely coded in Fortran and, although a powerful tool in terms of performance, was difficult to use and to modify.

The *IceChrono* dating model (Parrenin et al., 2015) was developed, based on the same principles as Datice, providing improvements and simplifications in the mathematical, numerical and programming aspects with respect to Datice. However, IceChrono is sometimes slower than Datice depending on the resolution chosen, in particular because IceChrono performs a numerical gradient calculation while Datice has a more efficient analytical gradient calculation. IceChrono is coded in Python 2.

Both Datice and IceChrono were initially developed for ice core studies. Technically, it is possible to use these dating tools to combine chronological information from other paleoclimatic archives by fixing the thinning function at unity and by discarding the air bubbles variable (Bazin et al., 2019), but the models are not designed for this purpose.

In this paper, we present a new probabilistic model, named *Paleochrono-1.1*, which is the successor of IceChrono. Paleochrono-1.1 is specifically designed to combine chronological information from multiple types of archives. Paleochrono-1.1 is coded in Python 3 and is more efficient, easier to use and more accurate than IceChrono. We first detail the Paleochrono-1.1 methodology (section 2) and demonstrate its utility with two dating experiments: one with only a speleothem and one that combines ice cores and speleothems (section 3). We then present the computing resources used by Paleochrono-1.1 in various dating experiments (section 4) and discuss our results (section 5).

## 2 Method

Paleochrono-1.1 is designed to work with paleoclimatic archives where it can be assumed that the deposition rate is strictly positive at any time. Paleochrono-1.1 does not detect hiatuses, working at a temporal resolution where the deposition process can be considered as continuous (because, for example, it does not snow every day on an ice sheet) but hiatuses can be included by treating the different continuous sections independently. We use the term *site*, to refer to one individual archive; we do not use the term record since it can mean a particular proxy within a site. We use *depth* in a broader sense to refer to the distance along the archive, from the youngest to the oldest section. The age along the archive is

therefore continuous and strictly increasing. If there is a known hiatus in the archive, the sections before and after the hiatus should be considered as two different sites in Paleochrono-1.1. If there is a reversed section in the archive (e.g. the section 3,320-3,345 m in the Vostok ice core, Raynaud et al., 2005), this section should be considered as a different site and its depth axis inverted.

Paleochrono-1.1 is set up for two types of archives: the so-called *simple archives*, with one unique material (and therefore only one depth-age relationship), constant density, and no post-depositional thinning (e.g., speleothems, marine sediments), and *ice-core archives*, where we deal with two materials (the ice and the enclosed air) with different age-depth relationships, variable density, and where post-depositional thinning occurs.

## 2.1 Method summary

The true chronology of a paleoclimate site is a function of the deposition rate. In the case of ice cores, two additional variables must also be considered: the LID of the air bubbles and the thinning function. These variables form what we call the forward model, and they are unknown. To find the optimal chronology, they must be estimated based on:

• Prior information about their values at each site;

• Chronological observations (Figure 1) such as: ages at certain depths, the time elapsed between two depths, the synchronicity of events recorded at two different sites that are assumed capable of recording events simultaneously, or the depth difference of air and ice levels of the same age within the same ice core ($\Delta_{depth}$).

All these different types of information, which can be mathematically described as probability density functions (PDF), are assumed to be independent and are combined together using a Bayesian framework to obtain posterior estimates of the input variables (deposition rate and, for an ice core, LID, and thinning) and of the resulting chronologies. Uncertainties on the prior estimates and on the observations are assumed to be Gaussian and the forward model is linearized, allowing this problem to be solved as a least-square optimization.

## 2.2 The forward model

### 2.2.1 For a simple archive

For a simple archive, the archiving model is:

$$\chi(z) = \chi_0 + \int_{z_0}^{z} \frac{dz'}{a(z')} \tag{1}$$

where $z$ is the depth along the paleoclimatic record, $\chi(z)$ is the age at depth $z$, $\chi_0$ is the age at the top depth $z = z_0$ (i.e. the age of the youngest material at the site), and $a$ is the depth-dependent deposition rate at the site of deposition.

This equation integrates along the depth axis the number of annual layers per unit depth from the surface.

### 2.2.2 For an ice-core archive

For an ice-core site, the archiving model is slightly more complicated, since we need to account for the age of the air inside the ice and for the vertical thinning of ice layers:

$$\chi(z) = \chi_0 + \int_{z_0}^{z} \frac{D(z')}{a(z')\tau(z')} dz', \tag{2}$$

$$\psi(z) = \chi(z - \Delta_{\text{depth}}(z)), \tag{3}$$

$$\int_{z-\Delta_{\text{depth}}(z)}^{z} \frac{D(z')}{\tau(z')} dz' = l(z) \times \left.\frac{D}{\tau}\right|_{firn}^{0}, \tag{4}$$

where $z$ is the depth along the ice core, $\chi(z)$ is the age of the ice at depth $z$, $\chi_0$ is the age of the ice at the top (with depth $z = z_0$), $a$ is the deposition (also called accumulation) rate along the ice core, $D$ is the

175 (dimensionless) density relative to pure ice, $\tau$ is the vertical thinning function (also dimensionless), $\psi$ is the age of the air, $\Delta_{\text{depth}}$ is the depth difference between air and ice of the same age, $l$ is the lock-in-depth of air bubbles and $\left.\frac{D}{\tau}\right|_{firn}^{0}$ is the average value of $\frac{D}{\tau}$ in the firn when the air particle was at the lock-in-depth (this parameter is usually ~0.7 for most firns, see Parrenin et al., 2012).

Equation 2 integrates along the depth axis the number of annual layers per unit depth from the surface. Equation 3 describes that the air age at depth $z$ is equal to the ice age at depth $z - \Delta_{\text{depth}}$, which is the definition of $\Delta_{\text{depth}}$. Equation 4 describes that if one corrects a depth interval between an ice depth and the depth of air of the same age for thinning, one gets the initial firn thickness in ice-equivalent units.

### 2.2.3 Numerical aspects

The parameters $a$, $\tau$ and $l$ are discretized onto a fine depth grid called the *age-equation grid*. Equation (1) (for a simple archive) or Equation (2) (for an ice-core archive) are solved using a cumulative sum. Then Equation (4) for the $\Delta_{\text{depth}}$ is solved to deduce the air age from Equation (3). To solve Equation (4), we first integrate $\frac{D}{\tau}$ from the surface down to every depth in the age-equation grid, i.e. we have a correspondence table between real depths and unthinned-ice-equivalent (UIE) depths. Then, for every actual air depth in the age-equation grid, we obtain the air UIE depth from the table. We then subtract the right-hand side of Equation (4) from this air UIE depth to get the ice UIE depth. Finally, we use the correspondence table to obtain the real ice depth and the $\Delta_{\text{depth}}$. When we need to compute the age of the ice or air or $\Delta_{\text{depth}}$ at depths which are not nodes of the age-equation grid (for example when comparing the model with observations, see below), we use a linear interpolation.

## 2.3 The probabilistic problem

### 2.3.1 General probabilistic consideration

The general idea of Paleochrono-1.1 is to combine different sources of chronological information: prior knowledge on the archiving process, together with various chronological observations (e.g. radiometric ages). The assumption is that each site and each site pair (used for stratigraphic links) have independent information. Moreover, within each site and within each site pair, the various types of chronological information are assumed to be independent. For example, for a given site, the prior scenario of the deposition rate, the dated horizons and the dated intervals are all assumed to be independent. It is only within each type of information for a certain site or a certain site pair that it is possible to define the correlation of the information. For prior archiving scenarios of deposition rate (and thinning and LID for

an ice core), we do this by defining correlation matrices which have a triangular form, that is, the correlation matrix has ones along its diagonal and the correlation linearly decreases to zero when the age difference (for the deposition rate and LID) or the depth difference (for the thinning function) reaches a user-defined $\lambda$ value. Setting these correlation matrices for the prior allows to have a weighting which does not depend on the resolution chosen for the inversion grids. Indeed, each interval of length $\lambda$ will have a weight of 1, which is the same weight as one observation. As a consequence, the cost function converges towards a single value when the resolution is increased.

In mathematical terms (Tarantola, 2005), combining different independent sources of information corresponds to multiplying the probability density functions (PDFs) of the prior and of the observations. The result of this multiplication is called the likelihood function. Here, we assume the PDFs to be independent multivariate Gaussian distributions. Multiplying the PDFs therefore corresponds to adding least squares terms.

### 2.3.2 The least-squares cost function

We write the cost function as follows:

$$J = \sum_k J_k + \sum_{k<m} J_{k,m},$$ (5)

where $J_k$ is the term related to site number $k$ and $J_{k,m}$ is the term related to site pair $(k,m)$.

For a simple archive, the cost function term is the sum of terms related to the age at the top of the sequence $\chi_0$, deposition prior information, dated horizons, and dated intervals. For an ice-core archive, the cost function term is the sum of terms related to the age at the top of the site $\chi_0$, the prior information (for deposition, thinning, and LID), dated horizons (in ice or air), dated intervals (in ice or air), and $\Delta_{\text{depth}}$ observations.

For a site pair of simple archives, the cost function term simply contains a term related to the stratigraphic links. For a site pair of ice-core archives, the cost function term is the sum of four terms related to ice-ice, ice-air, air-ice and air-air stratigraphic links. For a mixed site pair of a simple archive and an ice-core archive, the cost function term is the sum of two terms related to ice and air stratigraphic links.

Each of additive components of the $J_k$ and $J_{k,m}$ terms described above is written as:

$$J = \boldsymbol{R}^T \, C^{-1} \, \boldsymbol{R} \tag{6}$$

where $\boldsymbol{R}$ is a residual vector, i.e. a vector containing the differences between the model values and the observations/prior values divided by the uncertainty of the observations/prior, and $\boldsymbol{C}$ is a correlation matrix. This correlation matrix can be Cholesky-decomposed as $\boldsymbol{C} = \boldsymbol{L}\,\boldsymbol{L}^T$, so that Equation (6) can be rewritten:

$$J = (\boldsymbol{L}^{-1}\boldsymbol{R})^T \, (\boldsymbol{L}^{-1}\boldsymbol{R}) \tag{7}$$

At the end, we therefore have the sum of independent scalar residuals.

### 2.3.3 The input variables

The general idea is to adjust the age of the top of the record and the deposition rate (as well as the thinning function and LID for an ice-core archive) so as to minimize the cost function. We call these variables the *inverted variables*. The prior estimates of deposition rate (and thinning and LID for an ice core) are transferred by interpolation onto the age-equation grid. A first approach could be to adjust these variables on the same age-equation grid, as is done in the Datice software (Lemieux-Dudon, 2009; Lemieux-Dudon et al., 2010b, a, 2015). This requires the inversion of many variables, hence considerable computing resources. Here we follow the same approach as in IceChrono (Parrenin et al., 2015), where we define a multiplicative correction function onto a coarser grid, called the *inversion grid*. These correction functions are defined as a function of the age for the deposition rate and LID and as a function of the depth for the thinning function. We then transfer these correction functions on the age-equation grid by a linear interpolation, using the prior age (the age calculated from the prior scenario of deposition, LID and thinning) for the deposition rate and the LID. A basic assumption of our method is that the inverted variables are always strictly positive, otherwise the Equations (1-4) have a singularity for a value of zero for deposition rate and thinning. The correction function should therefore stay strictly positive as well. Such variables are called *Jeffreys variables* and are generally described by log-normal distributions (Tarantola, 2005). Through the application of a change of variable using the logarithm function, Jeffreys variables become Cartesian variables and are then described by Gaussian probabilities.

### 2.3.4 The optimization method

This least-squares problem is solved iteratively using a *trust region* algorithm (Branch et al., 1999; Byrd et al., 1988), which converges toward a minimum of the cost function and stops when a convergence criterion is met (the default value is $10^{-5}$ but it can easily be changed). At each iteration, the model is linearized, that is, we consider the linear operator which is tangent to the model. This linear system is solved and the solution is then used as the base of the next iteration.

Contrary to IceChrono, the Jacobian operator is calculated analytically. For each site and each site pair, the Jacobian matrix is derived for the residuals. For additional efficiency, the linear tangent operator and its adjoint are derived for the multi-site logic. Indeed, when using the linear solver, one does not need to calculate the Jacobian matrix, but just the effect of this matrix or its transpose on a vector. The linear tangent and adjoint operators are therefore an efficient way to solve a least-squares problem, especially when the Jacobian matrix is sparse because this Jacobian matrix does not need to be fully formed.

The initial value of the input vectors is set either as the prior, randomly or from the result of a previous dating experiment. The latter option is possible even if a previous experiment did not have the same depth or age resolutions. It is therefore possible to use a bootstrap method, starting from a fast, low-resolution experiment to spin and following up with a high-resolution experiment.

The trust region algorithm provides optimized values of the input vectors. At the solution, the product of the transpose Jacobian matrix with itself gives an approximate value of the Hessian matrix, that is, the inverse of the posterior covariance matrix. From there, the covariance matrix of the input variables for each site is calculated. Then, the covariance matrix for the output variables for each site is calculated.

### 2.3.5 Choosing the statistical parameters of the prior

At this stage, it may seem difficult and subjective to determine the statistical parameters of the prior, namely the uncertainties and the correlation lengths. There are two possible approaches.

In the first approach, the prior is defined from general knowledge on the archive. Typically, we may know from previous experiences on other sites that our archiving model is correct within a certain uncertainty level. For example, for ice cores, our deposition model is generally good within 10-20%, but for

speleothem, the deposition is highly variable and our model with a constant deposition rate might be in error by a factor 2-5 for some sites and some time periods.

In the second approach, following the principle of Occam's razor, the uncertainty of the prior is defined as the simplest possible model which reasonably fits the observations, providing there are enough observations to constraint the statistical model. In this approach, the uncertainty (resp. correlation length) is decreased (resp. increased) as much as possible while still keeping an acceptable agreement between the posterior model and the observations. We propose to study the residuals of the different type of

observations (corresponding to each term of the cost function) after the optimization process to decide if the values for these variables were chosen correctly. To help interpret the residuals, their distributions are fitted with a Student-t distribution. We propose to tune the uncertainties of the prior to have a scale ~0.2-0.5, that is, the observations are generally fitted well enough. We then tune the correlation lengths to have a number of degrees of freedom ($N_{DF}$) as large as possible (that is, distributions with small tails). Having

small tails means there are no problematic observations which are contradictory with the prior.

### 2.3.6 Detection of outliers

Paleochrono-1.1 also detects possible outliers in the observations, which indicate some incompatible chronological information given to the model. If a given observation is not fitted by the model within a given tolerance level (by default this level is $3\sigma$), a warning is displayed at the end of the run and points

directly to this observation. Of course, the incompatible information can sometimes be due either to a wrong observation or to an overestimation of the confidence on a prior constraint. Thus, the user has to decide between these two possible explanations: either the user can remove the observation, or increase the flexibility of the prior to fit this observation.

### 2.4 Programming aspects

Paleochrono-1.1 is coded in the Python 3 programming language with several scientific packages (numpy, scipy, matplotlib). Paleochrono-1.1 both solves the optimization problem and displays the results as figures. For a simple archive, the figures show deposition rate and age. For an ice-core archive, the figures depict deposition rate, thinning function, LID, ice age, air age, $\Delta_{depth}$, $\Delta_{age}$, and ice layer thickness.

Paleochrono-1.1 also displays the distribution of various residuals as histograms, in particular for each site, each site pair and each type of observation. These residuals are fitted with a Student-t distribution, whose parameters, centre location (loc), scale and number of degrees of freedom ($N_{DF}$) are given in the figure. This helps the user define the right values for the error bars and correlation lengths.

We use the trust region algorithm as implemented in the `scipy.optimize.least_squares` function. Compared to IceChrono, several coding optimizations have also been made.

The core of the code is entirely separated from the dating experiment directory which also contains the results of the run and which is composed of general parameter files, a directory for each site (which contains the parameters and observations for the given site) and a directory for each site pair (which contains the observations for the given site pair). Parameters that are common to all sites or site pairs can be set directly in a single file. All the parameter files follow the YAML format (Paleochrono-1.1 uses the pyaml module), which, contrary to IceChrono, allows one to make a clear separation between the main code and the parameters. The input data are given in text files. With respect to IceChrono, all files are given more straightforward names. It is not necessary to understand Python to run the code. Documentation on how to use Paleochrono-1.1 is available within the code. The output figures have also been improved with respect to IceChrono.

There are outputs at each steps of the run, including the initialisations, the optimization, the computing of the confidence intervals and the construction of the graphs, so that the user can inspect the process and estimate how much time is needed for completion. In particular, during the optimization by the trust region algorithm, the value of the cost function and its reduction is displayed at each iteration. After the optimization, possible outliers are detected in the observations.

There are no fixed units in the code, so it is possible to use any unit system, but the units must be consistent. For example, it is possible to use kyr as the age unit instead of yr, but then this unit has to be used everywhere. Also, for a particular site, the depth can be expressed in another unit system (e.g., mm for speleothems), but then the deposition rate is expressed using the derived unit (e.g., mm/yr for speleothems).

## 3. Example dating experiments

### 3.1 Dating of the Hulu MSL speleothem

To first demonstrate the ability of Paleochrono to date simple archives like speleothems, we date the MSL speleothem from Hulu Cave (Wang et al., 2001) and compare the resulting age model with age models derived by other methods/software, as given in the SISALv2 database (Comas-Bru et al., 2020). There are four age models used in this database: Bchron (Haslett and Parnell, 2008), Bacon (Blaauw and Christen, 2011), copRa (Breitenbach et al., 2012) and StalAge (Scholz and Hoffmann, 2011).

We applied Paleochrono with a depth grid between 6 and 450 mm with a step of 1 mm. The deposition grid is defined between 30 kyr and 80 kyr with a step of 100 yr. We chose values of 100% for the deposition rate uncertainty (that is, the deposition rate is allowed to vary by an exponential factor) and 1000 yr for the correlation length to allow for millenial scale variations. This parametrisation accounts for the highly irregular deposition process of speleothems.

We show the result of the Paleochrono age model on Figure 2, together with the SISALv2 age models and the dated horizons used in all age models. Paleochrono generally reproduces consistent age-depth relationships with respect to the other age models. The posterior uncertainties are smaller than the ones obtained with Bchron and Bacon, but comparable with the copRa and StalAge uncertainties.

### 3.2 The AICC2023-Hulu dating experiment

IceChrono was used on the AICC2012 dating experiment (Parrenin et al., 2015), a chronology that combines the EPICA Dome C (EDC), Vostok (VK), Talos Dome (TALDICE) and EPICA Dronning Maud Land (EDML) Antarctic ice cores and the NorthGRIP (NGRIP) Greenland ice core (Bazin et al., 2013; Veres et al., 2013).

Paleochrono-1.1 was also used on the AICC2023 dating experiment (Bouchet et al., 2023), an incremental improvement of AICC2012, with updated prior scenarios and dated horizons for the EDC ice core over the last 800 kyr. For this experiment, we increased the resolution until the cost function converges towards a single value. This value is ~196, which is small with respect to the number of observations (2,139), highlighting the good general match of the posterior scenario with respect to the observations.

Here, to test and demonstrate the ability of Paleochrono-1.1 to date simple archives in combination with ice cores, we incorporate into this AICC2023 experiment two last glacial speleothems (MSD and MSL) from Hulu Cave (Wang et al., 2001), that have U/Th dated horizons (Cheng et al., 2018) on a 18-55 ka BP time interval. We caution readers that this dating experiment has been constructed to test the

Paleochrono-1.1 model and to illustrate its abilities, and as such the resulting chronology presented here is *not* intended to be used for paleoclimatic studies. A future effort to update the AICC2023 chronology with information from speleothems is planned, but this is beyond the scope of the current study.

We set the prior deposition scenario to constant for both speleothems with a relative uncertainty (1σ) set to 1 (that is, the deposition is allowed to vary by an exponential factor). We assume also a correlation

length of 1,000 yr for the deposition rate of both speleothems, allowing for millenial scale variations in the deposition rate. For each speleothem, we use published U/Th dated horizons (Cheng et al., 2018). Then, we define stratigraphic links between (1) the NGRIP $\delta^{18}O_{ice}$ and the MSD $\delta^{18}O_{calcite}$, (2) the MSD and MSL $\delta^{18}O_{calcite}$ and (3) the MSL $\delta^{18}O_{calcite}$ and the EDC $CH_4$ records. To do so, we take advantage of the fact that the abrupt climate variability characterising the last glacial period (Corrick et al., 2020;

NorthGRIP project members, 2004) is clearly expressed in each of these records. We link the records at the onset of each abrupt Dansgaard-Oeschger (DO) events (Figure 3) using a mid-slope approach by assuming a global synchroneity in the timing of the rapid warming transitions in ice cores and of the $\delta^{18}O_{calcite}$ changes in speleothems (Adolphi et al., 2018; Corrick et al., 2020). We assign a constant uncertainty (1σ) of 100 yr to these synchronisation horizons. 100 yr is a rough estimate of the

synchronisation error during DO transitions based on the duration of the transition in the different archives (Capron et al., 2021; Corrick et al., 2020). A more careful analysis would be needed to refine these estimates individually, but this is beyond the scope of the current manuscript. In AICC2023, the layer-counting GICC05 constraint (Svensson et al., 2008) was used as dated horizons with small (<50 yr) uncertainties. This choice was made to maintain a compatibility with GICC05 but does not correspond to

the true information the layer counting provides. Here, we choose instead to use the constraint from GICC05 as dated intervals of 1000 yr durations, assuming no correlation in counting errors.

We call this experiment *AICC2023-Hulu*. Figure 3 shows the NGRIP $\delta^{18}O_{ice}$ record, Hulu/MSD and Hulu/MSL $\delta^{18}O_{calcite}$ records and EDC $CH_4$ record onto this common and optimized age scale. Figure 4

and Figure 5 are automatically generated by the Paleochrono-1.1 software and represent the chronology and deposition rate for the MSL speleothem as well as the synchronisation for the NGRIP-MSD site pair, respectively.

We can observe in Figure 4 that the posterior chronology for the MSL speleothem is in better agreement with the dated horizons than the prior chronology (which is expected since a constant growth rate is assumed in the prior case) and generally fits the U-Th ages within their confidence interval. Paleochrono-1.1 is also able to interpolate in-between age horizons when they are less dense. The uncertainty of the posterior chronology ranges from 50 to 400 yr (1σ), increasing when the age horizons are less dense or less precise.

Figure 5 shows that Paleochrono-1.1 is able to reconstruct a variable deposition rate from the chronological information, in particular the dated horizons along the MSL speleothem. It is also able to estimate an uncertainty on this posterior reconstruction, which will depend mainly on the uncertainty of the U/Th dated horizons, the depth resolution of the U/Th dates and the assumed growth-rate variation that affects interpolation uncertainty.

In Figure 6, we observe that Paleochrono-1.1 starts with a prior scenario based on archiving models with asynchronous stratigraphic links and that this new tool is able to come up with a solution where these stratigraphic links are respected.

# 4. Benchmarks

We now test the computing performance of Paleochrono-1.1 (computing time, memory used) in various dating experiments and configurations. We use a computing server with a Bi pro Xeon 2.10 GHz (48 cores and 96 threads) and 256 Gb of RAM. We use the Anaconda distribution of Python, which contains the Intel MKL (Math Kernel Library), providing parallel algorithms for common mathematical operations.

## 4.1 Comparison to IceChrono

We perform a test of computing performances on the AICC2012-VHR (Very High Resolution) experiment as defined in Parrenin et al. (2015). In this experiment, there are 10,520 variables to be inverted and 1,939 observations.

The experiment took about 24h and used 4.8 GB of RAM using IceChrono. The same experiment took 2 mins and used 3.6 GB of RAM using Paleochrono-1.1. This is a factor ~700 difference in computing time and 25% less memory. This improvement in terms of computing resources will allow users to increase the number of sites in a dating experiment, increase the resolution within a paleoclimatic site, and/or run the software on their desktop computer rather than on a server.

## 4.2 Computing time as a function of the number of computing cores

We now test the Paleochrono-1.1 software with different numbers of computing cores. We use the AICC2023 dating experiment as defined in section 3. We test with 1, 2, 4, 6, 8, 12, 16, 24, 32 and 48 computing cores. Figure 7 shows the result of this experiment. The total computing time (resp. optimization time) ranges from 3 m 50 s (resp. 1 m 51 s) for 48 computing cores, to 24 mn (resp. 2 m 55 s) for one computing core. In this dating experiment, the Paleochrono-1.1 software therefore scales quite well from one core up to 16 cores (with a scaling factor of ~6), but after 16 cores the computing time stagnates. In this experiment, 16 therefore seems to be the optimal number of computing cores, but this result might depend on the number of inverted parameters in the dating experiment.

## 4.3 Resources as a function of the number of inverted parameters

We now test the resources required by Paleochrono-1.1 as a function of the number of inverted parameters. We use the AICC2023 dating experiment (Bouchet et al., 2023). Starting from a low-resolution experiment, we multiply the resolution of deposition rate, LID and thinning in this experiment by factors of: 1, 2, 5 and 10. As a result, the number of inverted parameters is: 6,478, 12,944, 32,332 and 64,644. The third experiment corresponds to the official AICC2023 experiment (Bouchet et al., 2023). The results are shown on Figure 8 (on a log scales). The computing time ranges from ~0.7 m to ~14.2 m. The memory used ranges from ~2.1 GB to ~107.3 GB. We can observe that the memory evolves roughly

quadratically as a function of the number of inverted parameters. For the computing time, the evolution is close to linear for a small number of parameters and then becomes quadratic for a large number of parameters.

## 5 Discussion and perspectives

### 5.1 On the use of probabilistic dating methods

First, we discuss why we should combine chronologies in the first place and what "optimal" means in this context. Common chronologies with internally-consistent sources of chronological information have the advantage to be more accurate because they incorporate more data and their uncertainties. A probabilistic method can capitalize on the fact that each dating method has its own strengths and they are complementing each other. In the example employed here, the speleothems provide very accurate absolute ages via radiometric U/Th dating, whereas the NGRIP ice core provides very accurate relative ages (i.e., durations) from counting of annual layers across intervals. Combining these two archives could therefore provide a chronology that is more accurate than the age scale of each archive dated in isolation. Moreover, common chronologies allow to decipher temporal sequence of climate and environmental changes between different archives as lead and lags are not originating from chronology differences (assuming the synchronization links are realistic).

Second, we enumerate several applications of probabilistic dating models. Datice, the predecessor of Paleochrono-1.1, was used to create a coherent chronology of the EDC, EDML and NGRIP ice cores during the last deglaciation (Lemieux-Dudon et al., 2010b). Later, Datice was used to build AICC2012 (Bazin et al., 2013; Veres et al., 2013), a common chronology for 4 Antarctic ice cores (EDC, VK, EDML, TALDICE) and one Greenland ice core (NGRIP), taking into account many different types of absolute or relative chronological constraints. AICC2012 was used to discuss the comparison and phasing of EDC, EDML, TALDICE and NGRIP during the DO events of the last deglaciation (Landais et al., 2015). Datice was also used in a multi-archives context to build a coherent chronological scenario in the Mediterranean region during the last deglaciation for 3 lake-sediments cores, 2 speleothems and one marine core (Bazin et al., 2019). More recently, Paleochrono-1.1 was used to build the DF2021 chronology for the Dome

Fuji ice core (Oyabu et al., 2022), taking into account dated horizons and Δdepth constraints from various

methods. Paleochrono-1.1 was also used to reconstruct the temporal variations of surface mass balance around Dome Fuji (Antarctica) for the last 5,000 years using shallow ice cores and snow pits (Oyabu et al., 2023) synchronized using volcanic horizons. The ST22 chronology of the Skytrain ice core was constructed by matching its stratigraphy to EDC through various ice and air records and by using Paleochrono-1.1 to obtain a best fit (Mulvaney et al., 2023). Finally, AICC2023, an update of AICC2012

with in-particular more accurate orbital-tuning records, was recently build using Paleochrono-1.1 (Bouchet et al., 2023).

Third, we discuss the limitation of probabilistic dating methods. Common chronologies are also attached to drawbacks: they may mask systematic differences between dating methods that should be investigated and solved rather than forced into the same framework. In this case, the optimal solution in probabilistic

terms may actually represent a compromise that is less physically meaningful. Additionally, the models for combining all the information may grow to be so comprehensive that most users will not be able to maintain an overview of the data employed, and operating the model will entail sometimes implicit and important choices, e.g. on how to estimate the error bars of the prior and of the observations, how to set the correlation lengths for the prior, etc.

Another limitation of the method is that it requires errors to be both independent and have Gaussian distribution. Radiocarbon calendar ages typically do not have a Gaussian uncertainty, as they are a convolution of the measurement uncertainty and a calibration curve taking into account the variable atmospheric $^{14}C$ history. U-Th ages close to the limit of the technique also have an asymmetrical uncertainty which cannot be considered as Gaussian. Likewise, volcanic ties have a complex uncertainty:

if the tie correctly links the same layer in two records, the uncertainty of the synchronization will usually be no more than a few years, determined by data resolution and the shape of the signal matched. However, if the tie is incorrect, the error can be any number, and the error will rarely be described well by a Gaussian distribution with a width of e.g. 200 years. Furthermore, errors in any type of age constraint will be correlated in case of systematic biases. For example, when a sequence of closely spaced volcanic layers

(forming a recognizable pattern due to their similar spacing) are matched between two ice cores, the tie points are likely all correct or all erroneous, making their uncertainties highly correlated. In the

experiment performed here, there may be temporal lags in the climate system between the various parameters that are stratigraphically linked, the annual layer counting could systematically over- or under-count layers beyond what is included in the counting uncertainties, and the U/Th dating may have systematic biases related to for example the detrital Th correction.

In conclusion, optimal chronologies are practical for users who want to use the best possible common chronology, but it absolutely does not replace the need to compare and improve the chronologies of individual sites. The compromises involved in the modelling entail a risk that wrong chronological information or insufficiently quantified uncertainties will influence the resulting time scale negatively in a non-transparent way.

## 5.2 Comparison with other dating softwares

Paleochrono-1.1 was applied to the MSL speleothem (Hulu Cave, Wang et al., 2001) and compared to the SISALv2 age models (Comas-Bru et al., 2020). Compared to other age models, the one obtained with Paleochrono-1.1 tends to be less conservative, with smaller uncertainties than age models obtained with Bchron (Haslett and Parnell, 2008) and Bacon (Blaauw and Christen, 2011) and comparable uncertainties with the age models obtained with copRa (Breitenbach et al., 2012) and StalAge (Scholz and Hoffmann, 2011). The resulting depth-age curve is also generally smoother, therefore implying less variations in the deposition rate.

Compared with other softwares, Paleochrono-1.1 assumes that the uncertainties are gaussian, therefore it cannot reproduce asymmetric or multimodal uncertainties. But it can manage very large experiments and multi-site experiments, which other software cannot currently do.

## 5.3 Comparison with IceChrono

Paleochrono-1.1 is an evolution of IceChrono (Parrenin et al., 2015), since it started as the same code base. But Paleochrono-1.1 improves several aspects of IceChrono. The first improvement to note is that Paleochrono-1.1 can handle continuous paleoclimatic archives other than ice cores. We showed here a test with two speleothems from Hulu Cave and multiple ice cores from Greenland and Antarctica. Other improvements include:

- **Efficiency:** Paleochrono-1.1 uses significantly fewer computing resources, in particular computing time. This is thanks to:

  - o the new trust region algorithm which can use an iterative solver at each linear iteration
  - o the per-site analytical Jacobian matrix
  - o the prior and multi-sites linear tangent and adjoint operators
  - o the ability to define the convergence criterion
  - o various code optimizations

  On a AICC2012-VHR dating experiment identical to the one shown in Parrenin et al. (2015), the gain in computation time is a factor ~700, which is huge. Therefore, Paleochrono-1.1 is more adapted for high-resolution runs than IceChrono. The memory reduction is less significant, with only a 25% reduction.

- **Ease of use:** Paleochrono-1.1 is easier to use than IceChrono thanks to several improvements:

  - o Paleochrono-1.1 now uses parameters files in the YAML format instead of the python format. This makes it easier to work with these parameter files and creates a clear separation between the main code and the parameters.
  - o Paleochrono-1.1 now uses a simpler naming of parameters and input/output files.
  - o Paleochrono-1.1 now has pre-defined grid types. Apart from the 'regular' type with a
  530 constant step, there is a 'quadratic' type with a linearly increasing step. It is also easy to add new grid types. This makes it easier to have, for example, age grids which are refined for present times and coarse for past times, or depths grids for ice cores which are more refined near the bedrock.
  - o There is an interactive output during the optimization process that displays the evolution
  of the cost function at each iteration. This makes it easier to inspect the optimization process and estimate the time that is still needed for the run to complete.
  - o The output figures have been greatly improved. In particular, the resolution is displayed on the deposition rate, LID and thinning figures. Moreover, it is possible to define the units used and the figures will respect these units.

• **Accuracy:** Paleochrono-1.1 is more accurate than IceChrono because of the following improvements:

   o Paleochrono-1.1 now inverts $\chi_0$, the age at the top of the sequence, while it was prescribed in IceChrono. This is an improvement since it is not always possible to accurately determine the age of the top of the sequence, in which case it needs to be estimated within

545 the optimization process.

   o The forward model is more accurate, thanks to several inaccuracies which have been corrected.

   o Paleochrono-1.1 does not estimate the thinning in the firn for Equation (4) but directly prescribes $\frac{D}{T}\Big|_{firn}^{0}$. This allows Equation (4) to be exact instead of being an approximation.

o Thanks to the analytical Jacobian, the optimization finds a better minimum, that is, in closer agreement with the observations.

In summary, Paleochrono-1.1 is a considerable improvement with respect to IceChrono and we highly recommend the use of Paleochrono-1.1 to any existing and interested users of IceChrono. Indeed, the future maintenance of IceChrono will be discontinued.

## 5.4 Computing resources

In the AICC2023 dating experiment, we have tested the computed time needed by Paleochrono-1.1 as a function of the number of computing cores used. We find that 16 seems to be the optimal number of computing cores, with a code which scales with a factor ~6 with respect to one unique computing core. We have successfully tested the Paleochrono-1.1 software in the AICC2023 experiment with ~65,000

inverted parameters and ~2,000 observations. We find that the resource requirement (computing time, memory) evolves roughly quadratically as a function of the number of inverted parameters for large dating experiments. In this experiment, the limiting factor is the memory. Indeed, the most demanding run takes only 14 mins, which is acceptable but the memory used (107 GB) is on the upper range of what can be found in a workstation. Using a computing node with ~ 1TB of memory, which is at the edge of what can

be done currently with shared memory, would allow one to invert for ~200,000 parameters.

## 5.5 Limitations and possible future improvements

Paleochrono-1.1 assumes that the correlation matrices of the prior have a triangular shape with a defined correlation length, that is, only a local correlation is considered. It could be possible to have more complex forms of correlation.

Paleochrono-1.1 only deals with continuous depth-age models. It could be interesting in the future to include possible hiatuses, with an automatic detection.

Regarding resources, the most memory-demanding part in this run is not the optimization itself, but rather the construction of the posterior covariance matrix, which is required to evaluate the uncertainty of the posterior scenario. Another possibility to decrease the memory used and further increase the number of inverted parameters would be to not form the whole posterior covariance matrix, but only subsets of it, as seems possible using the LSQR linear solver (Kostina et al., 2009).

Another limitation of Paleochrono-1.1 relates to the subjective choice of the statistical parameters of the prior, in particular the $\sigma$ (uncertainty) and $\lambda$ (correlation length) parameters. We explained how we can estimate them iteratively, but it would be better to optimize these statistical parameters at the same time as the physical parameters of the model.

For a multi-sites experiment, Paleochrono-1.1 requires to manually define synchronization links between the different sites based on the visual resemblance of records. An automatic synchronization method within Paleochrono-1.1 would allow to circumvent this time-consuming and subjective step and would allow to account for the archiving constraint at the same time.

Paleochrono-1.1 only deals with Gaussian uncertainties and assumes the forward model is almost linear in the vicinity of the solution of the optimization problem. Going beyond these assumptions would require to solve the optimization problem with a Monte Carlo Markov Chain (MCMC) process. Such method would allow to deal with more general probability distribution functions (PDF), but at the expense of larger required resources for a given problem. Therefore, large problems could not be solved using this method.

# Conclusions

Here we describe a new probabilistic dating model for continuous climate archives, called *Paleochrono-1.1*. It is an evolution of the IceChrono model originally dedicated to ice cores, but it can now handle other continuous climate archives based on a varying deposition rate, such as lake and marine sediment cores or speleothems, and produce a coherent age scale between these multiple records and archives. Paleochrono-1.1 is more efficient, easier to use and more accurate than IceChrono. We demonstrate the ability of Paleochrono-1.1 on the MSL speleothem from Hulu Cave alone, and compare the resulting age model with the SISALv2 age models. As a second application, we apply Paleochrono-1.1 in an AICC2023-Hulu dating experiment where we add two radiometrically-dated speleothems from Hulu Cave to the AICC2023 dating experiment (Bouchet et al., 2023). We then benchmark the computing resources needed to run Paleochrono-1.1.

Paleochrono-1.1 has already been used in several published studies, either on one unique ice-core site but with multiple age constrains (Oyabu et al., 2022) or with multiple ice-core sites and multiple age constrains (Bouchet et al., 2023; Mulvaney et al., 2023; Oyabu et al., 2023). Many other applications of Paleochrono-1.1 would be possible for various time periods and using various paleoclimatic records.

# Code availability

Paleochrono-1.1 is an open source model available under the MIT license. It is hosted on the github facility (https://github.com/parrenin/paleochrono) and the version corresponding to the submission of this manuscript has been published on Zenodo (https://doi.org/10.5281/zenodo.10580279). The main author (F. Parrenin) can provide support for people wanting to use the software.

# Author contribution

FP developed the model code. MB, AL, KK, IO and RM tested the software. FP ran the simulations which were discussed between all co-authors. FP prepared the manuscript with contributions from all co-authors.

# Competing interests

The authors declare they have no conflict of interest.

# Acknowledgements.

We are thankful to Anders Svensson for helpful discussions. We thank Lucie Bazin for helpful discussions and for testing the Paleochrono-1.1 software. This work was supported by the Fondation Ars et Cuttoli CO2Role project and by the LEFE IceChrono and CO2Role projects. Some preliminary computations presented in this paper were performed using the GRICAD infrastructure (https://gricad.univ-grenoble-alpes.fr), which is supported by Grenoble research communities. EC acknowledges the financial support from the French National Research Agency under the "Programme d'Investissements d'Avenir" through the Make Our Planet Great Again HOTCLIM project (ANR-19-MPGA-0001). SOR gratefully acknowledges the support from the Carlsberg Foundation to the ChronoClimate project. CB acknowledges financial support from the US National Science Foundation (grants 1643394 and 1702920). This publication was generated in the frame of Beyond EPICA. The project has received funding from the European Union's Horizon 2020 research and innovation programme under grant agreement No. 730258 (Oldest Ice) and No. 815384 (Oldest Ice Core). It is supported by national partners and funding agencies in Belgium, Denmark, France, Germany, Italy, Norway, Sweden, Switzerland, The Netherlands and the United Kingdom. Logistic support is mainly provided by ENEA and IPEV through the Concordia Station system. The opinions expressed and arguments employed herein do not necessarily reflect the official views of the European Union funding agency or other national funding bodies. This is Beyond EPICA publication number XX.

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

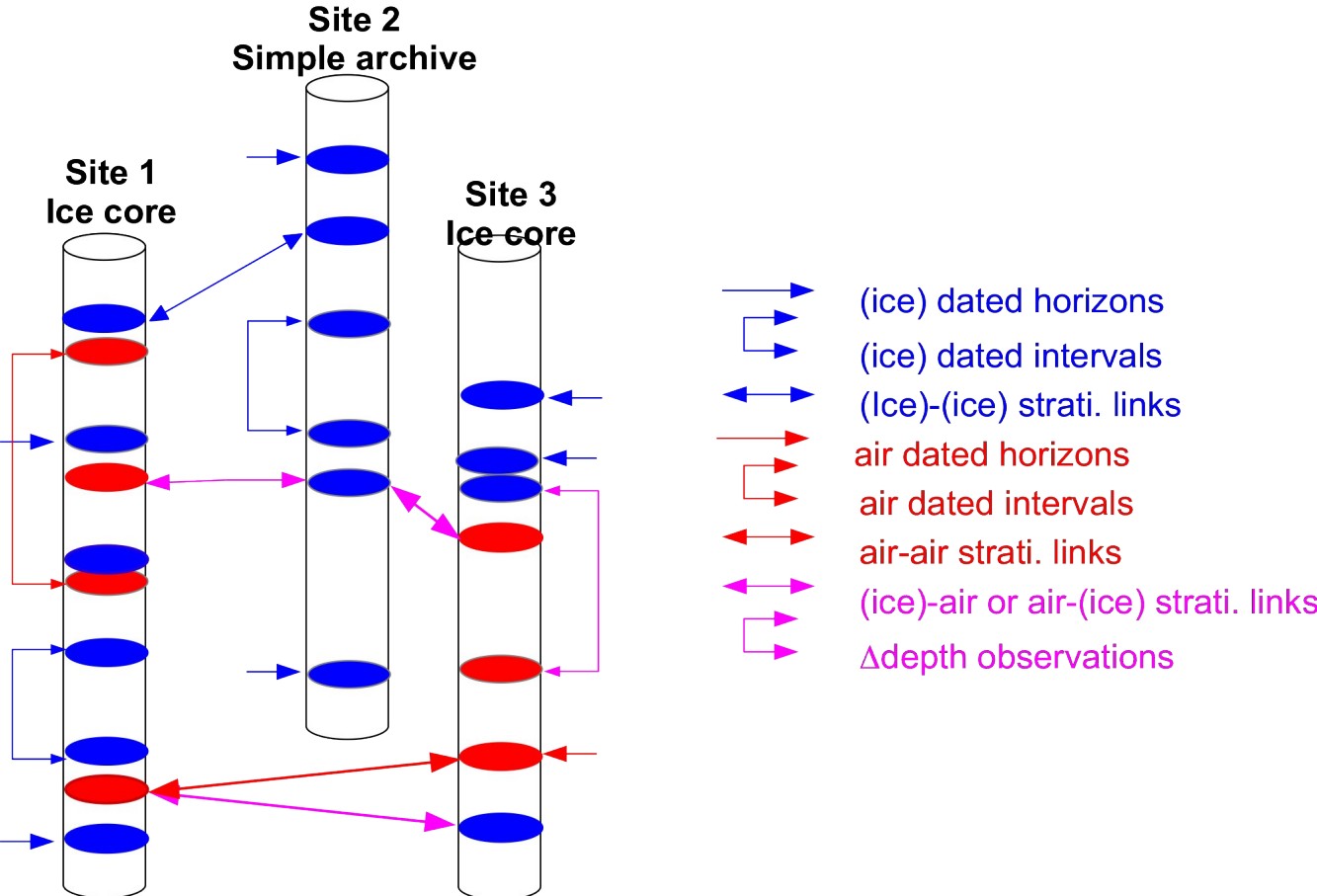

Figure 1: Scheme illustrating the different kinds of observations used to constrain the chronologies of the paleoclimatic sites in the Paleochrono-1.1 probabilistic dating model. The blue colour refers to the primary material (ice for an ice core), while the red colour refers to the secondary material (air for an ice core). The pink colour refers to mixed information involving the primary and secondary materials. In the legend, the term "ice" is in-between parentheses, since for a simple archive (e.g. such as a sediment core or a speleothem), there is no need to specify the material which is unique.

820

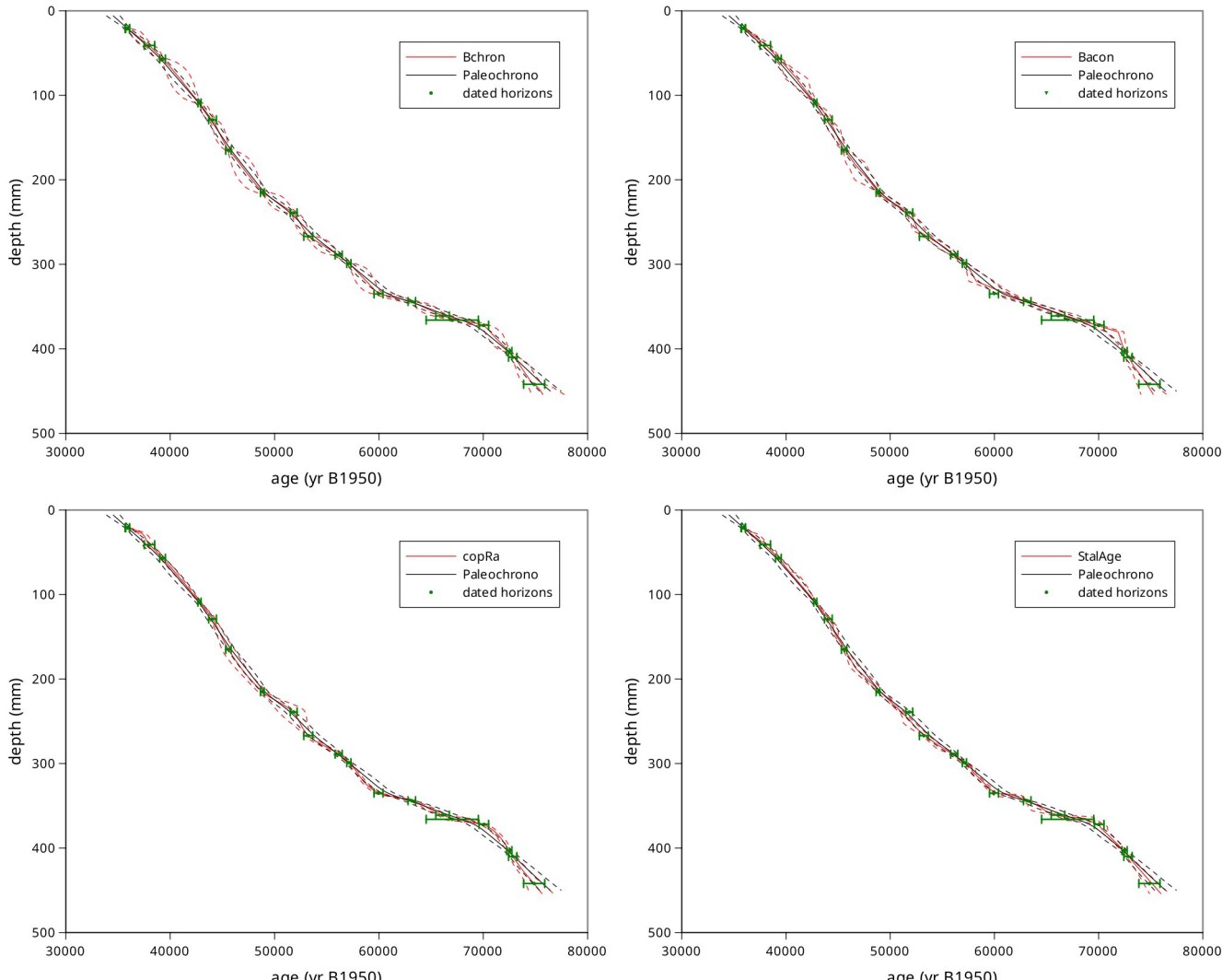

**Figure 2: Comparison of age models for the MSL speleothem (Hulu Cave, Wang et al., 2001). The Paleochrono age model is in black, the SISALv2 age models are in red and the dated horizons are in green. Top-left: Bchron (Haslett and Parnell, 2008). Top-right: Bacon (Blaauw and Christen, 2011). Bottom-left: copRa (Breitenbach et al., 2012). Bottom-right: StalAge (Scholz and Hoffmann, 2011). For better visibility, 2-sigma errors are shown in this figure.**

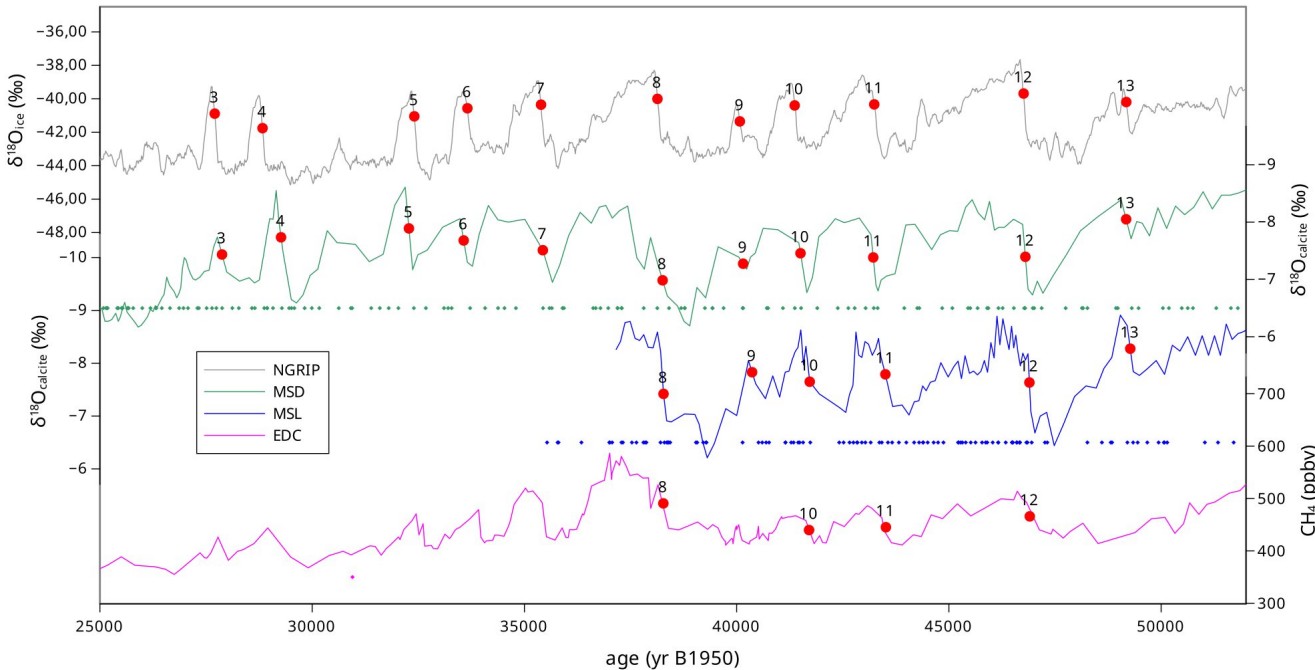

**Figure 3: Illustration of the synchronized and optimized chronology AICC2023-Hulu. From Top to bottom: NGRIP $\delta^{18}O_{ice}$ record; Hulu/MSD and Hulu/MSL $\delta^{18}O_{calcite}$ records; EDC $CH_4$ record. The red dots indicate the mid-points of DO onsets where the stratigraphic links are placed for the NGRIP-MSD, MSD-MSL and MSL-EDC site pairs and the labels indicate the DO numbers. The diamonds at the bottom of each panel represent the dated horizons used for each site.**

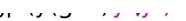

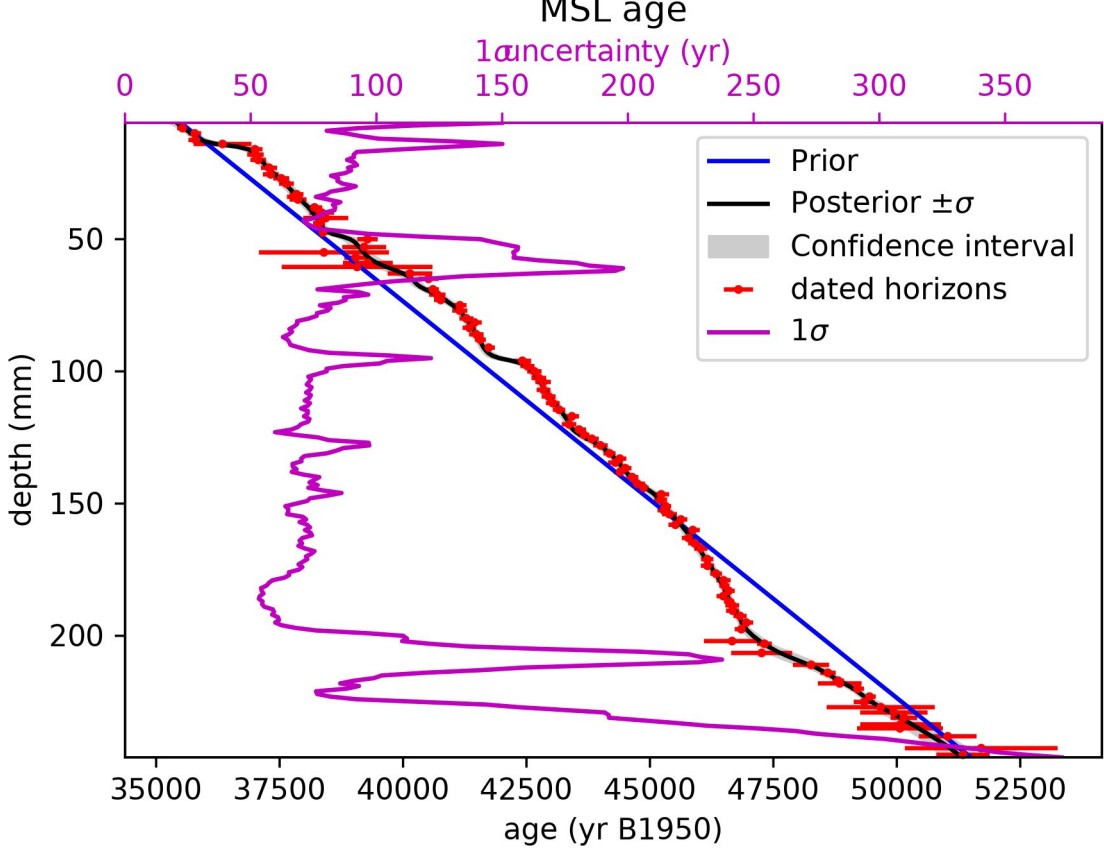

Figure 4: Age graph for the MSL speleothem as produced by Paleochrono-1.1 in the AICC2023-Hulu dating experiment. Blue: prior chronology based on the sedimentation scenario. Black and grey: posterior chronology and its confidence interval after optimization by Paleochrono-1.1. Red: dated horizons used in the dating experiment. Pink: 1-sigma uncertainty of the posterior chronology.

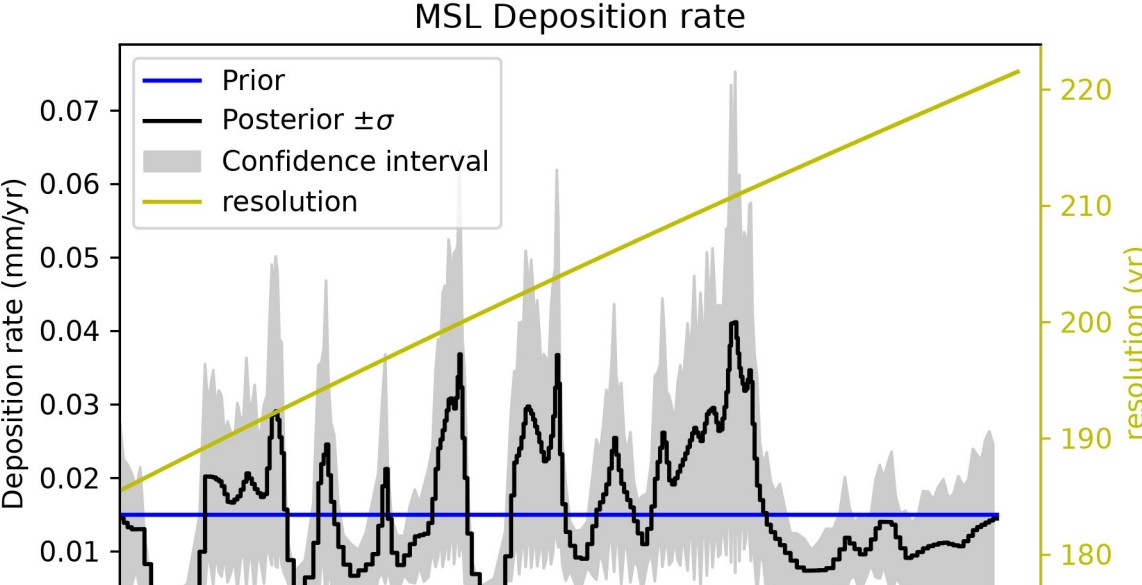

**Figure 5: Deposition/growth rate graph for the MSL speleothem as produced by Paleochrono-1.1 in the AICC2023-Hulu dating experiment. Blue: prior scenario. Black and grey: posterior scenario and its confidence interval after optimization by Paleochrono-1.1. Yellow: time resolution of the deposition rate in the inversion grid.**

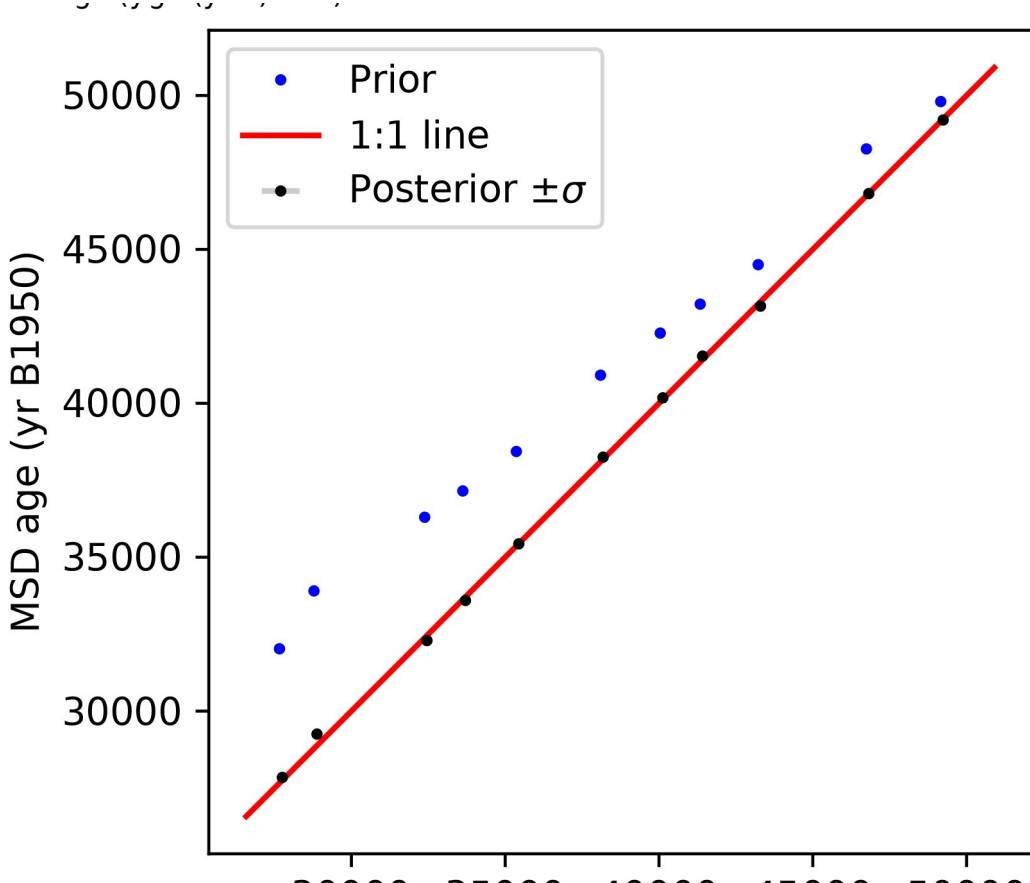

**Figure 6: Synchronisation graph for the NGRIP-MSD site pair as produced by Paleochrono-1.1 in the AICC2023-Hulu dating experiment. Blue: prior scenario. Black: posterior scenario after optimization by Paleochrono-1.1. Red: the 1:1 line for comparison.**

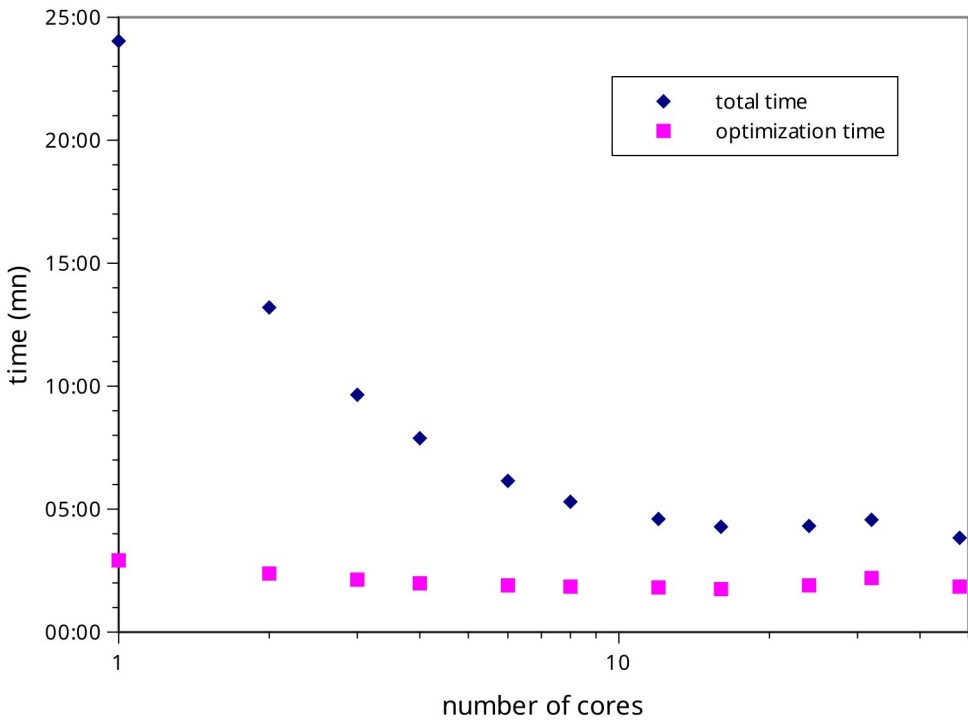

**Figure 7: Computing time of the AICC2023 dating experiment using Paleochrono-1.1, as a function of the number of computing cores.**

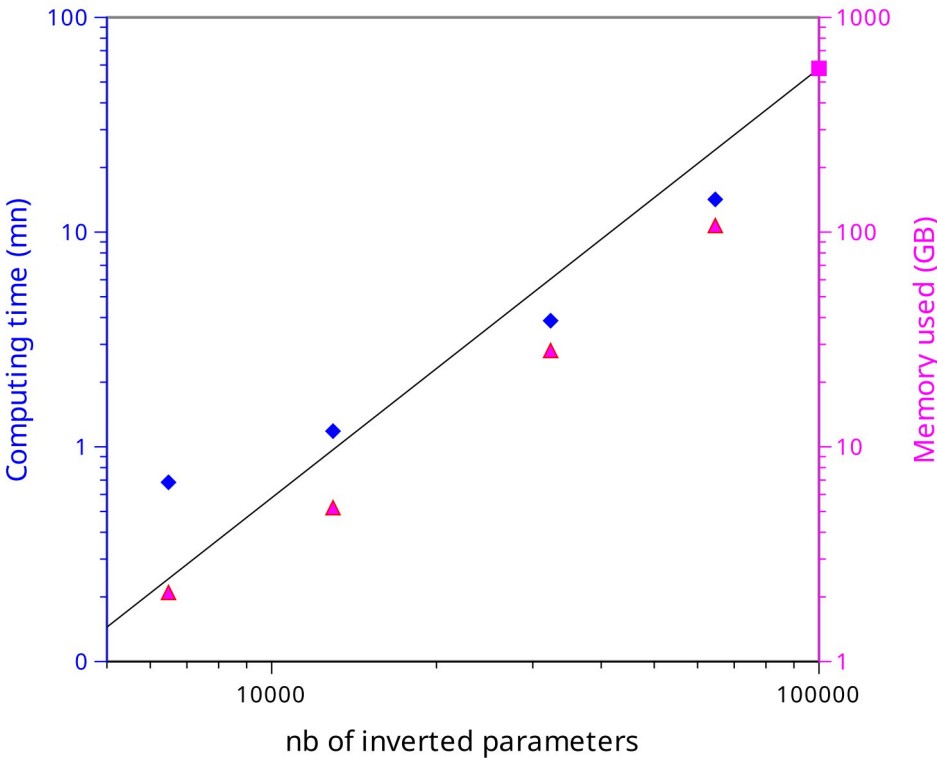

**Figure 8: Computing time and memory used by Paleochrono-1.1 in the AICC2023 dating experiment, as a function of the number of inverted parameters. The X- and Y-axes are logarithmic. The black line represents a quadratic evolution of the resources with respect to the number of parameters.**
