# Peer review of "The Paleochrono-1.1 probabilistic model to derive a common age model for several paleoclimatic sites using absolute and relative dating constraints"

_EGUsphere, 2023_

## Author Comment (AC1)

Review of the manuscript "The Paleochrono-1.1 probabilistic model to derive optimized and consistent chronologies for several paleoclimatic sites" by Parrenin et al., submitted to Geoscientific Model Development

General comment:

This is an interesting, well-written paper presenting a new probabilistic model (Paleochrono-1.1) to derive a common and optimised chronology for several paleoclimatic sites with potentially different types of archives. This is an important modelling question, which definitely fits within the scope of GMD. The new model builds on a previous model (IceChrono), but has substantial advantages compared with its predecessor (computational efficiency, ease of use, accuracy), which are clearly outlined and discussed in the paper, and can thus be considered as a substantial advance in terms of age modelling (in particular to combine different types of information from several archives).

All methods and assumptions appear valid and are clearly outlined, and the detailed description enables the reader to reproduce the results. The results of the example presented in the paper nicely demonstrate the general potential of the model. All conclusions are supported by the results.

[Figure]
 ¶

We sincerely thank the reviewer for his/her careful and relevant review, which helped us improve our manuscript.¶

The authors mention several aspects of the model (e.g., that it can also be used to construct age models for individual sites, the risk of choosing incorrect values for error bars and correlation lengths, etc.). The paper (and in particular the non-expert readers, who should use the model later) would strongly benefit from a more detailed explanation how to choose these values and their effect on the modelling results. Thus, I recommend to either present more examples or to calculate the same example with different choices of the parameters (see below for further details).

[Figure]
 ¶

This is a very relevant and important comment that deserves a careful answer.
Our probabilistic model derives from the theory of inverse problems (see, e.g. the books by Albert Tarantola). In this theory, the inverse method optimizes the physical parameters of the model given some probabilistic/statistical parameters (e.g., in our case, the values of the error bars and of the correlation lengths of the prior, the error bars of the observations). In the case of Paleochrono, while the observation error bars can be generally obtained based on external statistics (e.g. reproductible measurements), the statistical property of the prior can be deduced from general knowledge of the archive or from the observations themselves. In the long term and for a next version of Paleochrono, we would like to allow to infer these statistical

parameters at the same time as the physical parameters using Bayesian inferences. For now, we propose an iterative approach to update the statistical parameters of the prior based on the statistics of the observation residuals. ¶

We added the following section in the method:¶

**2.3.5 Choosing the statistical parameters of the prior*¶**

*At this stage, it may seem difficult and subjective to determine the statistical parameters of the prior, namely the uncertainties and the correlation lengths. There are two possible approaches.* ¶

*In the first approach, the prior is defined from general knowledge on the archive. Typically, we may know from previous experiences on other sites that our archiving model is correct within a certain uncertainty level. For example, for ice cores, our deposition model is generally good within 10-20%, but for speleothem, the deposition is highly variable and our model with a constant deposition rate might be in error by a factor 2-5 for some sites and some time periods.*¶

*In the second approach, following the principle of Occam's razor, the uncertainty of the prior is defined as the simplest possible model which reasonably fits the observations, providing there are enough observations to constraint the statistical model. In this approach, the uncertainty (resp. correlation length) is decreased (resp. increased) as much as possible while still keeping an acceptable agreement between the posterior model and the observations. We propose to study the residuals of the different type of observations (corresponding to each term of the cost function) after the optimization process to decide if the values for these variables were chosen correctly. To help interpret the residuals, their distributions are fitted with a Student-t distribution. We propose to tune the uncertainties of the prior to have a scale ~0.2-0.5, that is, the observations are generally fitted well enough. We then tune the correlation lengths to have a number of degrees of freedom ($N_{DF}$) as large as possible (that is, distributions with small tails). Having small tails means there are no problematic observations which are contradictory with the prior.*¶

In summary, I highly recommend publication in GMD, but I am convinced that the paleoclimate community would benefit from a more detailed illustration based on several examples. Below, I also list some other minor comments that may be useful to further improve the paper.

Detailed comments:

It may be good to modify the title to better illustrate the potential of the model to the general reader. The current title – in my opinion – does not illustrate that the model derives a common, combined age model from several sites using stratigraphic links, etc.

¶

We propose the following modified title:

The Paleochrono-1.1 probabilistic model to derive a common age model for several paleoclimatic sites combining absolute and relative dating constraints

Line 49: I would not consider speleothems as an archive with a continuous deposition process. We meanwhile know that many speleothems show various hiatuses (ranging from a few years to tens or hundreds of ka), and their growth thus rather needs to be considered as episodic. Later, it becomes obvious that the authors are aware of this, but it may be useful to state this major difference to, e.g., ice cores right at the beginning.

¶

Yes, good point.
We propose to add the following sentence in the introduction:

…and speleothems (Wang et al., 2001; Cheng et al., 2018; Corrick et al., 2020). **In the case of speleothem, however, we should note that the deposition (i.e. speleothem growth) somtimes is only episodic, that is continuous only during some time intervals.**

Line 58 ff.: Maybe mention the various, very sophisticated methods used in dendrochronology here.

We propose to mention dendrochronology in the 'intervals of known duration' bullet:

> *1) Intervals of known duration: sometimes, a section of an archive is of known duration (typically, a section **from an ice core or tree** where annual layers can be counted), although the absolute age of the section may not be known accurately.*

Line 127: "Paleochrono-1.1 does not integrate information regarding hiatuses, …" Even if this is true, as mentioned further down, hiatuses can be included: "If there is a known hiatus in the archive, the sections before and after the hiatus should be considered as two different sites in Paleochrono-1.1." This means that hiatuses will not be detected by the model, but they can be included. This should be clarified and may be very important for archives like speleothems (see above).

¶

We propose to modify this sentence as follow:

*Paleochrono-1.1 does not **detect** hiatuses, working at a temporal resolution **where the deposition process can be considered as continuous** (because, for example, it does not snow everyday on an ice sheet) **but hiatuses can be included by treating the different continuous sections independently.***

Line 134: "… simple archives, with one unique depth age relationship …" This is not clear to me. What does unique mean in this context. Please clarify.

¶

We propose to clarify this sentence as follow:

*Paleochrono-1.1 is set up for two types of archives: the so-called simple archives, with one unique **material (and therefore only one depth-age relationship)**, constant density, and no post-depositional thinning (e.g., speleothems, marine sediments), and ice-core archives, where we deal with two **materials (the ice and the enclosed air) with different age-depth relationships**, variable density, and where post-depositional thinning occurs.¶*

Line 151 ff.: "Uncertainties on the prior estimates and on the observations are assumed to be Gaussian …" This may be problematic for old (i.e., > 200 ka) U-series ages, where the errors become asymmetric. This should be included later, when, e.g., non-Gaussian uncertainties of 14C ages are mentioned.

¶

We propose to add the following sentence in section 5.1:¶

*Radiocarbon calendar ages typically do not have a Gaussian uncertainty, as they are a convolution of the measurement uncertainty and a calibration curve taking into account the variable atmospheric $^{14}C$ history. **U-Th ages close to the limit of the technique also have an asymmetrical uncertainty which cannot be considered as Gaussian.**¶*

Line 308 ff.: "We assume also a correlation length of 1,000 yr for the deposition rate of both speleothems, assuming higher frequency variations are absent." This is OK for this paper, but it may be noteworthy that the Asian speleothem d18O records often show a correlation on the precession time scale. Would this have an effect of the results?

If the prior sedimentation rate is flat, it will pull a bit the posterior towards a flat scenario, but it of course depends on the uncertainty one assigns to the prior. Of course, it would be possible to set up a correlation matrix which increases every precession cycle.¶

This is what we wrote in the perspective section:¶

***Paleochrono-1.1 assumes that the correlation matrices of the prior have a triangular shape with a defined correlation length, that is, only a local correlation is considered. It could be possible to have more complex forms of correlation.***¶

Line 317 ff.: "We assign a constant uncertainty (1σ) of 100 yr to these synchronisation horizons. 100 yr is a rough estimate of the synchronisation error during DO transitions." This is not clear to me. What is the effect of this uncertainty in the end? What would happen if the chosen value was too small? It may be interesting to demonstrate and discuss the effect for different values. Alternatively, more information should be provided to assist the readers how to choose this value.

¶

This uncertainty is determined from the duration of the DO transitions in the different archives and what is possible in terms of sychronisation. If the uncertainty is too small, it will try to overfit this information, leading possibly to unrealistic sedimentation scenarios. We propose to modify the sentence as follow:¶

*We assign a constant uncertainty (1σ) of 100 yr to these synchronisation horizons. 100 yr is a rough estimate of the synchronisation error during DO transitions **based on the duration of the transition in the different archives (Capron et al., 2021; Corrick et al., 2020)**.¶*

Line 331 ff.: "Figure 4 shows that Paleochrono-1.1 is able to reconstruct a variable deposition rate from the chronological information, in particular the dated horizons along the MSL speleothem. It is also able to estimate an uncertainty on this posterior reconstruction, which will depend mainly on the uncertainty of the U/Th dated horizons, the depth resolution of the U/Th dates and the assumed growth-rate variation that affects interpolation uncertainty." It is clear to me that the main scope of the paper is to demonstrate that the model can generate a common, combined model for different archives. However, since the model can also be used to calculate individual age models for single archives (e.g., speleothems) and will probably also be used for this purpose, it may be good to present the results for this as well. If so, it would be interesting how the results compare with other published age models (for speleothems, see, for instance, Comas-Bru et al., 2020).

Thank you for this interesting comment. We now produce a MSL dating experiment with the SISALv2 dated horizons and compare it with the 4 SISALv2 age models for this speleothem. We now added Figure 2 and section 3.1:

[Figure]

*Figure 2: Comparison of age models for the MSL speleothem (Hulu Cave, Wang et al., 2001). The Paleochrono age model is in black, the SISALv2 age models are in red and the dated horizons are in green. Top-left: Bchron (Haslett and Parnell, 2008). Top-right: Bacon (Blaauw and Christen, 2011). Bottom-left: copRa (Breitenbach et al., 2012). Bottom-right: StalAge (Scholz and Hoffmann, 2011). For better visibility, 2-sigma errors are shown in this figure.¶*

**3.1 Dating of the Hulu MSL speleothem¶**

*To first demonstrate the ability of Paleochrono to date simple archive like speleothems, we date the MSL speleothem from Hulu Cave (Wang et al., 2001) and compare the resulting age model with age models derived by other methods/softwares, as given in the SISALv2 database (Comas-Bru et al., 2020). There are four age models used in this database: Bchron (Haslett and Parnell, 2008), Bacon (Blaauw and Christen, 2011), copRa (Breitenbach et al., 2012) and StalAge (Scholz and Hoffmann, 2011).¶*

*We applied Paleochrono with a depth grid between 6 and 450 mm with a step of 1 mm. The deposition grid is defined between 30 kyr and 80 kyr with a step of 100 yr. By analysing the distribution of the residuals, we found optimal values of 30% for the deposition rate uncertainty and 1000 yr for the correlation length.¶*

*We show the result of the Paleochro age model on Figure 2, together with the SISALv2 age models and the dated horizons used in all age models. Paleochrono generally reproduces smoother age-depth relationships than the other age models. The posterior uncertainties are smaller than the ones obtained with Bchron, Bacon and copRa, but comparable with the StalAge uncertainties. The StalAge age model is the closest to the Paleochrono age model, although StalAge is not exactly as smooth as Paleochrono.¶*

We also added section 5.2 in the discussion:¶

**5.2 Comparison with other dating softwares¶**

*Paleochrono-1.1 was applied to the MSL speleothem (Hulu Cave, Wang et al., 2001) and compared to the SISALv2 age models (Comas-Bru et al., 2020). Compared to other age models, the one obtained with Paleochrono-1.1 tends to be less conservative, with smaller uncertainties than age models obtained with Bchron (Haslett and Parnell, 2008) and Bacon (Blaauw and Christen, 2011) and comparable uncertainties with the age models obtained with copRa (Breitenbach et al., 2012) and StalAge (Scholz and Hoffmann, 2011). The resulting depth-age curve is also generally smoother, therefore implying less variations in the deposition rate. ¶*

*Compared with other softwares, Paleochrono-1.1 assumes that the uncertainties are gaussian, therefore it cannot reproduce asymmetric or multimodal uncertainties. But it can manage very large experiments and multi-site experiments, which other software cannot currently do.*

Line 381 ff.: "… whereas the NGRIP ice core provides very accurate relative ages (i.e., durations) from counting of annual layers across intervals." Even if the relative accuracy of such layer counted chronologies is very high, the counting uncertainty sums up to considerably (e.g., for GICC05). How is this included in the model?

If provided with dated intervals from the layer counting, Paleochrono does reproduce absolute ages with errors that increase with depth because the layer counting errors sum up. This is dealt with in Paleochrono. Below is a figure of an experiment with just the NGRIP site and with only the layer counting information provided as dated intervals of 1,000 yr (the green rectangles), assuming no correlation of counting errors. You can see how the errors sum up in the total uncertainty which increases with depth (the pink curve).¶

[Figure]

Based on your comment, we decided to slightly modify the AICC2023-Hulu dating experiment:¶

***In AICC2023, the layer-counting GICC05 constraint (Svensson et al., 2008) was used as dated horizons with small (<50 yr) uncertainties. This choice was made to maintain a compatibility with GICC05 but does not correspond to the true information the layer counting provides. Here, we choose instead to use the constraint from GICC05 as dated intervals of 1000 yr durations, assuming no correlation in counting errors***¶

Line 407 ff.: "Additionally, the models for combining all the information may grow to be so comprehensive that most users will not be able to maintain an overview of the data employed, and operating the model will entail sometimes implicit and important choices, e.g. on how to estimate the error bars of the prior and of the observations, how to set the correlation lengths for the prior, etc." I completely agree with that and would like to encourage the authors to demonstrate the effect of, e.g., incorrect choices for error bars and correlation lengths of the prior. This would not only be helpful to avoid such mistakes, but also improve the applicability of the model.

¶

We now propose a section on how to choose the prior iteratively (see before).
¶

Line 427 ff.: "In conclusion, optimal chronologies are practical for users who want to use the best possible common chronology, but it absolutely does not replace the need to compare and

improve the chronologies of individual sites. The compromises involved in the modelling entail a risk that wrong chronological information or insufficiently quantified uncertainties will influence the resulting time scale negatively in a non-transparent way." This is a very important point, and - again – I think, it would be very useful to better demonstrate the mentioned effects in the paper. One way would be to (i) construct an individual age model for one of the speleothems (and compare with one of the published models in SISAL), then (ii) construct the common model, and then (iii) construct a (iii) model using wrong chronological information or insufficiently quantified uncertainties to demonstrate the effect.¶

So we did (i) (see comment before) and we did (ii). For (iii), we have improved Paleochrono to detect outliers and we wrote a section in the method:¶

**2.3.6 Detection of outliers*¶**

*Paleochrono-1.1 also detects possible outliers in the observations, which indicate some incompatible chronological information given to the model. If a given observation is not fitted by the model within a given tolerance level (by default this level is 3σ), a warning is displayed at the end of the run and points directly to this observation. Of course, the incompatible information can sometimes be due either to a wrong observation or to an overestimation of the confidence on a prior constraint. Thus, the user has to decide between these two possible explanations: either the user can remove the observation, or increase the flexibility of the prior to fit this observation.*¶

Conclusions: This is mainly a repetition of the previous sections and could be shortened.

¶

We removed from the conclusion the part on the resources, which are technical and not of interest to the general reader.¶

Comas-Bru, L., Rehfeld, K., Roesch, C., Amirnezhad-Mozhdehi, S., Harrison, S.P., Atsawawanunt, K., Ahmad, S.M., Ait Brahim, Y., Baker, A., Bosomworth, M., Breitenbach, S.F.M., Burstyn, Y., Columbu, A., Deininger, M., Demény, A., Dixon, B., Fohlmeister, J., Hatvani, I.G., Hu, J., Kaushal, N., Kern, Z., Labuhn, I., Lechleiter, F.A., Lorrey, A., Martrat, B., Novello, V.F., Oster, J., Pérez-Mejías, C., Scholz, D., Scroxton, N., Sinha, N., Ward, B.M., Warken, S., Zhang, H. and SISAL Working Group Members (2020) SISALv2: A comprehensive speleothem isotope database with multiple age-depth models. Earth System Science Data 12, 2579–2606.

---

## Author Comment (AC2)

This paper describes the Paleochrono (1.1) model, which is intended to allow the construction of consistent age models for different sites and archives. This is likely to be a very important tool, particularly for the ice core community but potentially for other palaeoclimate communities. It carries out an important task that has not been accessible in an available program before. The methodology is carefully described, and appears logical, even if one could questions some aspects of the way errors are combined. The program's application is illustrated with a nice cross-archive example, and the computing performance is clearly described. Overall, I found this an important paper that should be published with only minor corrections.

Most of my comments are very minor single word clarifications. I have just two broader issues to raise.

¶

We warmly thank Pr Eric Wolff for his careful review of our manuscript.¶

In lines 310-320, the mid-point of DO transitions is synchronised with an uncertainty. This rests on the assumption that there are no or minimal lags between DO onsets in different sites and archives. For speleothems from different regions this is precisely what was shown by Corrick et al 202, using U/Th dates from different speleothems. However while they inferred it as likely, Corrick et al did not specifically demonstrate synchroneity between DO events in spelothems and in Greenland ice (or methane in Antarctic ice). This was rather done in Adolphi et al 2018. This illustrates a point that needs to be made more generally: that it is only OK to use tie points between archives if there is an a priori reason (mechanism (volcanic eruption), independently verified dates (U/Th dates in spelothems), or linkages (cosmogenic nuclide wiggle matching between ice and speleo)) to assume they are synchronous, and if the limit of synchroneity is specified (as it is at 100 years for speleo-ice at DO events). I know the authors know this but I think it needs spelling out, and the justification from the Adolphi and Corrick papers emphasised, to avoid the danger that genuine asynchroneity is falsely ignored by future users.

¶

Yes, we agree.
Paleochrono can take into account stratigraphic links in-between sites, but it is up to the user to check these links have been proven to link synchronous events.
We propose to clarify the description of our example AICC2023-Hulu dating experiment with the following modifications:¶

We link the records at the onset of each abrupt Dansgaard-Oeschger (DO) events (Figure 2) using a mid-slope approach by assuming a global synchroneity in the timing of the rapid warming transitions **in ice cores and of the $\delta^{18}O_{calcite}$ changes in speleothems (Adolphi et al., 2018; Corrick et al., 2020).**¶

I cannot claim to understand the details of how the cost function is calculated. I understand that each marker or correlation comes with an uncertainty. However I would have appreciated some discussion of how the priors are weighted compared to the age markers etc. Possibly this is a parameter inside the model?

The λ parameter sets the weighting for the prior, since any interval with a length of λ has a weight of 1. Any observation also has a weight of 1. We propose to specify this in section 2.3.1:

*Setting these correlation matrices for the prior allows to have a weighting which does not depend on the resolution chosen for the inversion grids. **Indeed, each interval of length λ will have a weight of 1, which is the same weight as one observation.** As a consequence, the cost function converges towards a single value when the resolution is increased.*

Minor comments

Line 67. "an event dated by radiometric analysis". Shouldn't this be a "layer dated by radiometric analysis"?

¶

Yes, layer is more precise, corrected.¶

Line 101. needs a ")" after "surface".

Thanks.

Line 133. Maybe mention reversals as well as hiatuses (reversals occur in deep Greenland ice for sure, and in some Antarctic ice).

Thanks for the comment. We added the following sentence:¶

***If there is a reversed section in the archive (e.g. the section 3,320-3,345 m in the Vostok ice core, Raynaud et al., 2005), this section should be considered as a different site and its depth axis should be inverted.*** ¶

Line 165. "$D$ is the (dimensionless) relative density of the snow/ice material". Clarify that this is relative to pure bubble free ice, not to (for instance) water.

¶

A relative density is always relative to the pure material, but sure we can make it even more clear:¶

…$D$ is the (dimensionless) **density relative to pure ice** of the snow/ice material…¶

Line 350 and line 487. Neither m nor mn seem like good abbreviations for minutes. I suggest spelling out or using "mins".

¶

Thanks, using "mins".¶

Line 400. I know you refer to it later but here would be a good place to reference Mulvaney et al 2023.

¶

Yes, sorry, this was missing, corrected.¶

Figure 1. I found the legend on the right confusing for the diouble-headed blue and purple arrows, because they are labelled as ice-air or ice-ice links, but could equally refer to speleo-air or speleo-ice links. I assume that was the meaning of putting ice in () but this is nowhere stated. It perhaps should be for clarity (ie add a statement that (ice) should be taken to refer to whatever sediment is used, eg speleothem or marine sediment as well as ice).¶

Yes, for a simple archive, there is only one material so no need to specify it. This is why we put "ice" in parentheses. We propose the following text in the legend to clarify Figure 1:¶

***The blue colour refers to the primary material (ice for an ice core), while the red colour refers to the secondary material (air for an ice core). The pink colour refers to mixed information involving the primary and secondary materials. In the legend, the term "ice" is in-between parentheses, since for a simple archive* (e.g. such as a sediment core  or a speleothem)*, there is no need to specify the material which is unique.***